# The Underestimated Role of Platelets in Severe Infection a Narrative Review

**DOI:** 10.3390/cells11030424

**Published:** 2022-01-26

**Authors:** Alberto Fogagnolo, Gianluca Calogero Campo, Matilde Mari, Graziella Pompei, Rita Pavasini, Carlo Alberto Volta, Savino Spadaro

**Affiliations:** 1Intensive Care Unit, Department of Translational Medicine and for Romagna, Azienda Ospedaliera Universitaria di Ferrara, University of Ferrara, 44121 Ferrara, Italy; matilde.mari@student.unife.it (M.M.); vlc@unife.it (C.A.V.); savinospadaro@gmail.com (S.S.); 2Cardiovascular Institute, Azienda Ospedaliero-Universitaria di Ferrara, 44124 Cona, Italy; cmpglc@unife.it (G.C.C.); graziella.pompei@outlook.it (G.P.); pvsrti@unife.it (R.P.); 3Maria Cecilia Hospital, GVM Care and Research, 48032 Cotignola, Italy

**Keywords:** platelet, platelet activation, infection, critical care

## Abstract

Beyond their role in hemostasis, platelets have emerged as key contributors in the immune response; accordingly, the occurrence of thrombocytopenia during sepsis/septic shock is a well-known risk factor of mortality and a marker of disease severity. Recently, some studies elucidated that the response of platelets to infections goes beyond a simple fall in platelets count; indeed, sepsis-induced thrombocytopenia can be associated with—or even anticipated by—several changes, including an altered morphological pattern, receptor expression and aggregation. Of note, alterations in platelet function and morphology can occur even with a normal platelet count and can modify, depending on the nature of the pathogen, the pattern of host response and the severity of the infection. The purpose of this review is to give an overview on the pathophysiological interaction between platelets and pathogens, as well as the clinical consequences of platelet dysregulation. Furthermore, we try to clarify how understanding the nature of platelet dysregulation may help to optimize the therapeutic approach.

## 1. Platelets Interactions with Bacteria

The cardiovascular system is usually a sterile environment; however, upon entry to the circulation, bacteria can interact with several cells leading to several complications including bacteremia, sepsis, infective endocarditis, disseminated intravascular coagulation and immune thrombocytopenia purpura. In all these conditions, a common feature is an abnormal platelet function caused by an interaction with the bacteria.

Although the main role of platelets is hemostasis, recently more attention has been focused on the role of platelets in the host response to infection [1,2,3,4,5]. However, under certain circumstances, the platelet response to infection may be a significant part of the problem.

Inflammation and thrombosis both contribute to reducing the spread of pathogens, and platelets migrate to the site of infection and detect pathogens, along with neutrophils [6]; in this manner, platelets migration prevents the dissemination of bacteria already located in the intravascular compartment. In addition to this, platelets use their protrusions at vascular microbreaches to prevent the invasion of extravascular bacteria [7]. This complex defense system contains bacterial spreading and promotes the elimination of bacteria from the circulation by sequestration in the hepatic and pulmonary vasculature. The complement system, which is involved in innate and acquired immune responses to different infections, is the fulcrum of the interplay between inflammation and thrombosis [6]. Some studies identified a shuttling mechanism in the spleen involving complement and platelet and concerning endovascular bacteria which allows to balance rapid clearance of pathogens with the induction of adaptive immune responses [8].

However, even if platelet response to infection is a crucial step in immune response, under certain circumstances it may be a significant part of the problem.

Bacteria can interact with platelets using different mechanisms; they may secrete products (as toxins) that bind to the platelet-causing activation independently of bacterial attachment [9,10] or may bind to platelets. The binding to platelets can be either a direct interaction or an indirect interaction. A direct interaction (Figure 1A) occurs when a bacterial adhesin binds directly to a platelet receptor [11,12]. An indirect interaction (Figure 1B) occurs when a bacterial adhesin binds to a plasma protein (or other soluble elements of the immune system such as immunoglobulins and complement proteins) which bridge the bacteria to a specific receptor on the platelet surface [11,12,13,14,15]. See Table 1 [16,17].

Bacteria can either promote platelet adhesion or can induce platelet aggregation. Platelet adhesion to bacteria is a measure of the strength of the interaction, whereas platelet aggregation is an indication of the quality of the interaction; in contrast to typical platelet aggregation induced by physiological agonists such as adenosine diphosphate (ADP), collagen or thrombin, bacteria induce an all or nothing response. In other words, there is a threshold concentration of bacteria, above which there is maximum aggregation and below which there is no aggregation at all [16,17]. Another specific feature of platelet aggregation induced by bacteria is “lag time,” which is a distinct pause in time before aggregation takes place. When a soluble agonist such as ADP is added to a suspension of platelets, the aggregation response occurs within a few seconds; otherwise, when bacteria are added to a platelet suspension, there is a concentration-dependent delay in the aggregation response [18]. Depending on bacteria, platelet aggregation may be preceded by a short lag time of around 2–5 min [11,19] or by a long lag time of about 12–18 min [13,20]. A short lag time usually indicates a direct interaction between the bacteria and the platelet, whereas a long lag time generally indicates an indirect interaction. The length of time relates to how long it takes for the bacteria to bind the bridging molecule and cross-react with the reciprocal receptor on the platelet [16].

The activation of platelets by bacteria can lead to different specific problems. If activation occurs in a localized manner, it can lead to thrombus formation; instead, a more systemic activation can lead to platelet consumption. Finally, activated platelets secrete many cytokines and other mediators that can trigger pathological processes. Infective endocarditis (typically due to Staph. aureus or an oral Streptococcus), in which a bacteria–platelet thrombus develops on the valve, is a typical example of a thrombotic complication of bacterial infection and can either lead to valve failure or the formation of a septic embolus [21]. During septicemia, platelet activation by systemic bacterial infection may lead to thrombocytopenia and bleeding complications due to platelet sequestration [22,23]; and this outcome relates to the entity of thrombocytopenia [24,25].

When activated, platelets secrete their granule contents, which contain at least 300 different proteins including cytokines and vascular activating factors [26,27]. These cytokines play a key role in the pathogenesis of atherosclerosis [26,28,29,30,31] and may also explain the association between infection and cardiovascular disease. As well as causing thrombocytopenia, sepsis also leads to shock due to endothelial inflammation and subsequent vascular leakage. Activated platelets play a key role in mediating endothelial damage [23,32,33].

Significant advances have been made in recent years in identifying the molecular mechanisms leading to platelet activation upon binding several bacterial species. Due to the rapid global emergence of multiple resistant strains of bacteria, it is critical that we identify novel drug targets that prevent unwanted platelet activation at the sites of injury in the vasculature. Given that different species of bacteria interact with platelets using various mechanisms, a correct interpretation of this phenomenon is crucial to develop future targeted therapies.

**Table 1 cells-11-00424-t001:** Bacterial–platelet interaction can vary depending on etiologic agent.

Platelet-Bacteria Interactions		
Direct adhesion	Indirect adhesion	
Short lag		Bridging protein
*S. sanguinis* [11]	*S. aureus* [12,13,14]	Fibrinogen
*S. aureus* [12,13,14]	*H. pylori* [19]	Fibronectin
		VWF
		Direct adhesion
Long lag		
*S. gordonii* [20]		
*S. sanguinis* [11]		
Non aggregating		
*S. gordonii* [20]	*H. pylori* [19]	

## 2. Platelets Interactions with Viruses

The relationship between activated platelets and immune response is maintained even when the infectious microorganisms are viruses [34,35,36,37]. Viral infection of cells begins with virus binding to a surface receptor that mediates its internalization, and platelets express various pattern recognition receptors (*PRR*) able to mediate binding and entry of various viruses [36,38,39,40]. The immune response against viruses is supported by consequent platelet activation and manifests itself through different mechanisms, such as the release of chemokines that promote endothelial signaling and leukocyte migration or by physically interacting with leukocytes [41,42,43]. Moreover, while traditional platelet activation by G-protein-coupled receptors is usually rapid, platelet *PRR* activation in responses to infections and immune stimuli can be delayed and sustained, persisting hours after initial aggregation and secretion [34,35,36,37,38,39,40,41,42,43,44].

Resultant virus-mediated thrombocytopenia is generally multifactorial; viruses use different strategies to decrease the levels of circulating platelets, either by decreasing platelet production or increasing platelet destruction. Platelets are produced in the bone marrow by megakaryocytes, and viruses can interfere with platelet production at various steps of development [36]. Some viruses, such as simian immunodeficiency virus (SIV) and human herpes virus, can influence the cytokine profile of the host, resulting in altered thrombopoietin (TPO) production in the liver [45,46,47]. Some others, including hepatitis C virus (HCV), also directly interfere with TPO production by destruction of liver tissue [48]. Moreover, human immunodeficiency virus (HIV), cytomegalovirus (CMV) and HCV replicate in megakaryocytes modulating their proliferation and function [49,50,51,52]. Nevertheless, thrombocytopenia induced by decreased platelet production is observed at later stages of infection; otherwise, rapidly induced thrombocytopenia during viral infections is mediated via enhanced platelet destruction. The most rapid way of platelet destruction occurs via direct interaction between platelets and viruses through a variety of receptors and it is mainly mediated by surface lectins, integrins and TLR [52,53]. Rotavirus utilizes the collagen receptor *GPIa/IIa* to bind to platelets [54,55], while hantavirus and adenoviruses interact with platelets via the fibrinogen receptor *GPIIb/IIIa*, the most abundant platelet integrin. [56] The Epstein–Barr virus’ (EBV) interaction with platelets occurs via complement receptor 2 (*CR2*) [57]; HIV and dengue virus activate platelets by binding to lectin receptors and a cell-specific intercellular adhesion molecule [58]. These direct interactions often result in platelet activation and adhesion of activated platelets to leukocytes. Platelet binding to neutrophils leads to phagocytosis of platelets and platelet activation itself stimulates platelet clearance in the liver and spleen [59,60]. However, platelets are not only activated by direct interactions with viruses. Host defense mechanisms in response to viral infections can also lead to platelet activation and decreases platelet life span [61]. Moreover, the B-lymphocyte production of antibodies against some viruses also interferes with platelet survival. These antibodies, which usually target surface glycoproteins of viruses, shows cross-reactivity with platelet surface integrins such as *GPIIb/IIIa*. That is called idiopathic thrombocytopenic purpura (ITP) or platelet autoantibody-induced thrombocytopenia and it has been described for HCV, HIV, CMV, EBV, hantavirus, varicella zoster virus, herpes viruses, and severe acute respiratory syndrome coronavirus [62]. The interaction between platelets and viruses is crucial even in the pathogenesis of the COVID-19 syndrome. Platelets from COVID-19 patients are more activated, aggregated faster and have increased expression of monocyte tissue factor [39,40]. Functional assays showed that platelets from COVID-19 patients are more responsive, sensitized to release inflammatory cytokines and adhere more efficiently [33]. All things considered, the data suggest that platelets may have the potential to contribute to the thrombo-inflammation in COVID-19 [33].

## 3. Cellular Changes in Platelet Structure and Function during Infection

The interplay between platelets and bacteria or viruses reported above can affect the structure and the function of platelets in several ways. The most known effect of infection is a fall in platelet count, which may show different levels of severity depending on the lowest level of platelet reached. The occurrence of thrombocytopenia has long been recognized as an independent risk factor for worst outcomes during infection, and the degree of thrombocytopenia is used as a marker of the severity during sepsis [33,63]. Indeed, the platelet count is included in the Sequential organ failure assessment (SOFA) score [64], the alteration of which is crucial in the diagnosis of sepsis.

Beyond the well-established prognostic finding of thrombocytopenia, there are other less-known platelet characteristics that may be evaluated during infection. Two indices of platelet morphology are easily available in most laboratories and have been shown to be affected by concurrence infections: the platelets distribution width (PDW) and the mean platelet volume (MPV). The PDW is a parameter of platelet heterogeneity, while MPV is a measurement of the average size of platelets. Platelet size is usually between 1.5 and 3 μm. Large platelets (3–7 μm) are called macrothrombocytes, whereas platelets reaching the size of erythrocytes or lymphocytes (larger than 7, up to 20 μm) are designated giant platelets [65]. Healthy subjects usually have less than 5% of large platelets, but infection-induced platelet activation is associated with major shape change due to cytoskeletal changes, including filopodial and lamellopodial extensions. These changes affect platelet size and variability and, therefore, MPV and PDW have recently been suggested as markers of platelet activation [66,67,68,69]. High PDW and MPV values were associated with 90-day mortality in patients with septic shock [62,63,64,65]; this was also in experimental animal models of endotoxemia [69]. Notably, studies conducted in non-infected critically ill patients, such as cardiac arrest [70], have not found any prognostic role of these morphological indices; these results suggest a direct role of infection on platelet morphologies, and a recent study confirmed this hypothesis [68]. As a result, both PDW and MPV can be used as a marker of infection severity and they are independent predictors of mortality during infections.

Outside the platelet morphological changes, several other biological changes can be described after platelet activation, which include the expression of different platelet receptors. P-selectin, which can bind leukocytes, is only expressed on the surface of activated platelets [71] and can be responsible for neutrophil-platelet aggregates in the circulating blood [72]. Similarly, CD40-Ligand (CD40L) is expressed on activated platelets, and can trigger an inflammatory response by interacting with CDBoth P-selectin and CD40L can be measured with a bead-based multiplex immunoassay and were associated with worse outcomes both in bacterial [73] and viral infections [74]. Remarkably, experimental studies showed compromised host defense to infection in P-selectin-deficient mice [75]; it appears that only a dysregulated platelet activation can be deleterious in patients with infections.

## 4. Main Techniques to Monitor Platelet Function

The in-vitro evaluation of platelet aggregation is used for the diagnosis of platelet function disorders [76]. There are many methods used to test platelet function; however, these are mainly for research purposes as a result of lack of standardization [77]. Light transmission aggregometry (LTA), developed in 1962 by Born and O’Brien, represents the gold standard for testing platelet function. This technology measures the changes in transmission of light through a sample of platelet-rich plasma (PRP) or platelet suspensions in buffer, which occur when platelets change shape and aggregate upon stimulation. Compared to other methodologies, LTA is far less influenced by platelet count [62]. However, LTA is a time-consuming and technically challenging technique and therefore is mainly used only in specialized laboratories. An alternative system to evaluate platelet aggregation in vitro is impedance aggregometry (IA). It consists of the calculation of the increase in electrical resistance between two electrodes immersed in a diluted sample of whole blood, PRP or platelet suspension, caused by the adhesion of platelets to the electrodes and the subsequent platelet aggregation [76,78]. The IA is calculated by the MultiplateTM analyzer [79], which relies only on platelet aggregometry; this might lead to misdiagnosing dense granule secretion defects. The lumi-aggregometer, a different version of LTA, provides information on platelet secretion in addition to platelet aggregation measures quantifying the ATP secretion with a luciferin/luciferase assay, while aggregation is assessed as in classical LTA. Despite several instruments available to measure lumi-aggregometry, few reports are available in the literature on its performance and validation [77]. To test the behavior of platelets in physiological conditions, it is necessary to add an element of shear in available assays (to mimic shear stress from the blood on the vessel). Between assays available, only the PFA-200^®^ requires blood to flow over a surface coated with a thrombogenic substrate and the assessment of platelets deposition and thrombus growth by microscopy [77]. Another factor that alters the evaluation of platelet function by traditional assays is low platelet counts. In patients with low platelet counts, the best option consists of flow cytometric assays of platelet activation markers. The use of flow cytometry has some advantages: a smaller volume of blood is needed without platelet-rich plasma preparation [77,80,81,82]. Several flow cytometry approaches have been successfully used in patients with severe chronic immune thrombocytopenia, showing that impaired platelet function is associated with bleeding, independent of platelet count. The reduced platelet count is related to altered paracrine amplification of platelet responses by ADP release. Therefore, platelet-count-adjusted reference ranges are needed [78,79]. The strengths and the limitations of the main techniques used to monitor platelet function are summoned in Table 2.

## 5. Platelet Response to Pharmacological and Non-Pharmacological Agents or Devices

Drugs represent the most common cause of platelet dysfunction. There are several agents used in intensive care units such as antibiotics, nonsteroidal anti-inflammatory drugs and volume expanders, which can impair platelet function. Between antibiotics, the compounds that most commonly affect platelet function are β-lactam antibiotics. Some of them produce predictable dose-dependent and duration-related effects on the bleeding time. Agonist-induced platelet aggregation is reduced from 25 to 75% in patients receiving large doses of β-lactam antibiotics. Their inhibitory effect is a maximum of one to three days after administration and can persist for several days after pharmacological discontinuance, suggesting that the action of antibiotics on platelets in vivo is presumably irreversible. Many mechanisms of alteration of platelets function have been proposed. Some antibiotics impair the interaction of platelet agonists (such as ADP and epinephrine) and/or von Willebrand factor with their corresponding receptors on the platelet surface. Antibiotics can also inhibit platelet function by binding to one or more membrane components necessary for adhesive interactions with the vessel wall [82,83,84].

Another complication induced using antibiotics is thrombocytopenia. Drug-induced immune thrombocytopenia can occur after common antibiotics administration like ceftriaxone, trimethoprim–sulfamethoxazole, vancomycin and penicillin. Typically, it occurs within one to two weeks of exposure to the drug, with an improvement within one to two days of drug discontinuation. Although rare, drug-induced immune thrombocytopenia can be fatal [85].

Nevertheless, in most cases, it is difficult to prove a causal relation of bleeding events to antibiotic therapy—particularly in intensive care units, where patients receive high doses of antibiotics and have multiple risk factors for hemorrhages.

Several pharmacological agents employed for their analgesic, anti-inflammatory or antipyretic effects can also impair platelet function by interfering with arachidonic acid metabolism. In apposition to aspirin, these drugs (including diclofenac, ibuprofen, indomethacin and naproxen) do not irreversibly block COX-1, but only as long as the drug circulates. Thus, their effect on platelets function is generally short lasting (<4 h) [86]. An exception is piroxicam, with a more prolonged platelet inhibition due to its plasma half-life of two days [84]. The individual bleeding risk induced by aspirin or other NSAIDs is quite unforeseeable (although often mild), but it rises significantly if concomitant comorbidities (e.g., hemophilia, chronic liver disease, renal failure), co-medication (e.g., antiviral drugs, anticoagulants) or some settings (e.g., delivery) predispose to hemorrhage [82,87].

Unfractionated heparin (UFH) and low-molecular-weight heparins (LMWHs) predispose to bleeding primarily through their anticoagulant effect, but they also have the potential to impair platelet function. Thrombocytopenia after heparin administration has been classified into two types: nonimmune heparin-induced thrombocytopenia and antibody-mediated heparin-induced thrombocytopenia, the latter commonly referred to as heparin-induced thrombocytopenia (HIT) [88]. HIT is an adverse drug reaction due to pathogenic antibodies against complexes of platelet factor-4 (PF4) and heparin. These antibodies activate platelets, neutrophils and monocytes, triggering platelet function, neutrophil extracellular trap formation and thrombin generation. Patients are unlikely to develop HIT after a short exposure to heparin because the pathologic antibodies form after a median of 4 days [89]. The risk of development of heparin-induced thrombocytopenia changes depending on the type of heparin, duration of exposure and the patient population, and it occurs in <0.1–7% of heparin-treated patients. HIT is more frequent after exposure to unfractionated heparin compared to low molecular weight heparin (LMWH). Among LMWHs, fondaparinux has a negligible risk of HIT and is increasingly used in the treatment of HIT. Treatment consists of stopping heparin administration and also switching to an alternative anticoagulant. Without treatment, the rate of thrombosis is about 6% per day [90,91]. Apart from that, heparins can intensify platelet responsiveness to weak stimuli such as ADP, associate directly with the platelet surface and promote platelet activation, as demonstrated by P-selectin expression and increased binding of fibrinogen or the fibrinogen-mimetic antibody PAC-1 to the platelet receptor αIIbβ3 [91]. Lately, it was shown that UFH promotes platelet responsiveness by potentiating αIIbβ3-mediated outside-in signaling. Where on the αIIbβ3 complex heparin specifically interacts is not known. Both abciximab and eptifibatide prevent platelet spreading on immobilized heparin suggesting that heparin may interact with αIIbβ3 [88].

Major surgery can cause hemostatic changes through different mechanisms related to surgical stress, tissue destruction and inflammatory reactions. These changes determine a shift of volume from the extravascular space to the intravascular compartment and interstitial space and thus hemodilution of coagulation proteins and an increased plasmatic fibrinogen concentration and platelets. These factors, combined with a hypofibrinolytic status, leads to a postoperative hypercoagulable state. Fluid infusion, in particular the administration of colloids, during surgery is associated with a prolongation of standard clotting times (CTs) and a decrease in plasma levels of coagulation factors and inhibitors of coagulation [92,93]. FVIII and von Willebrand factor (VWF) show a gradual increase during surgery—probably secondary to increases in stress hormones, including epinephrine and vasopressin—to inflammatory responses and endothelial activation. Moreover, the number of platelets increases and is not related to the degree of dilution, but presumably to the release of platelets from the spleen, lungs and bone marrow. Overall, major surgery is responsible for the perioperative prothrombotic state, with elevated levels of FVIII, fibrinogen, thrombin–thrombomodulin (TAT) complexes, VWF and hyperactive platelets. In the postoperative period, an increase of type 1 plasminogen activator inhibitor (PAI-1) levels also occurs [94,95]. Zhao et al. designed a study to assess the effect of bleeding speed on coagulation function during surgery. A total of 141 patients with massive bleeding undergoing pulmonary surgery were enrolled to compare the indicators of coagulation. The study showed how quick and slow bleedings have different impacts on the coagulation function, with quick bleeding causing poorer coagulation function in the short term, since changes of coagulation factors occur earlier than those of PLT in functional disorders because of surgical bleeding. Indeed, quick bleeding requires important amounts of coagulation factors in a short time to respond rapidly [96].

Blood-contacting medical devices (BCMDs) are frequently employed in device-assisted circulation to treat, replace or support diseased organs in human cardiovascular, pulmonary and renal systems. Despite BCMDs being an accepted therapy in these standard clinical practices, BCMD-related complications remain a significant challenge and are related to increased morbidity and mortality. The most common adverse events associated with BCMD-assisted circulation are thromboembolism and bleeding [97]. High shear stress caused by the contact between platelets surface and mechanical circulatory support devices has also been shown to cause impairment of platelet function. The origin of platelet dysfunction in patients undergoing mechanical circulatory support is likely multifactorial (contact with foreign surfaces, platelet activation and inflammatory and coagulative cascade activation) [98]. For example, hemorrhagic and thromboembolic complications are common during treatment with extracorporeal membrane oxygenation (ECMO), resulting in considerable morbidity and mortality. The most common complication is bleeding, occurring in 29–33% of adult ECMO patients. Thromboembolic complications are generally less common [99,100]. Thrombocytopenia is frequent in ECMO patients, regardless of the type of ECMO mode, and platelet count usually continues to decrease over the first 2–3 days up to seven days after the device implantation. Despite strong evidence still lacking, possible causes for a quick decline are concomitant of a patient’s primary disease, toxic drug effects and anticoagulation administration [98]. Between possible mechanisms responsible for platelet dysfunction, the contact of platelet artificial surfaces leads to platelet adhesion and activation, and high shear stress causes enhanced platelet activation, which may determinate an increased thrombotic propensity. At the same time, high shear stress can also cause the loss of platelet surface receptors important for platelet adhesion [98]. These alterations may result in impaired platelet adhesion and impaired activation, and thus in an increased risk of bleeding enhanced by the ongoing consumption of platelets. Antithrombotic therapy is necessary to maintain the patency of the ECMO circuit and to reduce thrombotic complications. Unfractionated heparin (UFH) is the most widely used anticoagulant, despite anticoagulation guidelines varying among ECMO centers. However, its use can result in a higher bleeding risk [99,100,101]. Reduced levels of platelets adhesion receptors for collagen and vWF (GPVI and GPIbα) and activation-dependent platelet surface markers (CD62 or P-selectin and CD63) are reported, compared with healthy individuals, during ECMO treatment [102,103]. A recent prospective observational study by Siegel et al. investigated platelet dysfunction and its relation to outcome in ECMO patients, showing platelets from ECMO patients as severely dysfunctional and thus predisposing patients to bleeding complications and poor outcomes. Compared to controls, the expression of platelet surface markers, delta granule secretion and formation of PLA was reduced, especially in response to stimulation. Baseline CD63 expression was higher and activated GPIIb/IIIa expression in response to stimulation was lower in non-survivors on day one of ECMO [104]. Putative causes of platelet dysfunction in patients undergoing ECMO are resumed in Figure 2.

Hemostatic imbalances are an issue also in critically ill patients requiring continuous renal replacement therapy (CRRT); even in this case, the contact of foreign surfaces with blood triggers procoagulant processes. Besides worsening the prothrombotic risk related to critical illness, even uremic platelet dysfunction is characterized by a bleeding tendency. A recent analysis using multiple electrode aggregometry (MEA) revealed an impaired platelet function affecting platelet activation via arachidonic acid, adenosine diphosphate, collagen and TRAP-6-related pathways at baseline in patients at surgical intensive care units (ICU) with acute kidney injury undergoing continuous veno-venous hemodialysis (CVVHD). Test results remained below defined reference ranges despite efficient elimination of urinary excreted substances [105]. Wand et al. investigated MEA performing MultiplateR in patients with acute kidney injury before the start of CRRT and 6, 12, 24 and 48 h after initiation of CRRT. Aggregometric analyses showed that arachidonic acid-induced platelet aggregation was significantly reduced after 6 h when compared to the baseline. The results of the present study indicate that CRRT may lead to impaired primary hemostasis, as shown by a decrease in ex vivo arachidonic acid-induced platelet aggregation [106].

Finally, the interaction between vaccines and platelets has recently gained much attention. Some cases reported the occurrence of immune thrombocytopenia and thrombosis syndrome after the administration of the ChAdOx1 nCoV-19 vaccine (AZD1222). The diagnosis of vaccine-induced immune thrombocytopenia and thrombosis (VITT) is based on specific clinical and laboratory data that should be used to distinguish by thrombosis with thrombocytopenia syndrome (TTS) [107]. The VITT disorder is associated with high-titer immunoglobulin G (IgG) class antibodies directed against the cationic platelet chemokine, platelet factor 4 (PF4) [108]. Adenovirus vaccine vectors can interact directly with and activate platelets, and these activated platelets are cleared from the circulation by Kupfer cells in the liver [109].

Most patients with thrombocytopenia and thrombotic events showed that increased levels of PF4 and anti-PF4 antibodies are found in patients with COVID-19 despite no history of heparin use [110]. Thus, vaccine-mediated thrombosis is due to immune complex formation. An interesting observation is that both SARS-CoV-2 infection and vaccination are both associated with thrombosis, although there are some clinical differences. Both infection and vaccination are associated with PF4-mediated thrombocytopenia. Conversely, while the infection leads to multi-organ failure, vaccination seems to lead to more focal thrombosis, typically in the brain or liver. Nowadays, the possibility of preventing and treating a rare event such as vaccine-related thrombosis remains an unresolved question.

## 6. Clinical Consequences of Platelets Dysregulation

As previously described, platelets play a pivotal role both in the regulation of coagulation and immune response in severe infection. Consequently, clinical effects of platelet dysregulation after severe infections can be categorized as (i) Thrombohemorrhagic complication (ii) reduced bacterial clearance (iii) endothelial damage and inflammation. Notably, the crosstalk between coagulation and immune response is confirmed by the fact that these different kinds of complications are usually linked [111,112]. The most described thrombohemorrhagic complication is the occurrence of sepsis-induced thrombocytopenia [63,113,114]. Nonetheless, even with a pseudo-normal platelet count, platelet activation can be responsible for organ failure throughout the formation of microthrombi from platelet aggregation. In a mouse model of peritonitis, the occurrence of platelet-rich thrombi in organ microvessels is correlated with organ failure [115]. The formation of platelet-rich thrombi can be enhanced by the high shear stress found in micro vessels during sepsis [116,117]. On the other hand, the interaction between platelets and leukocytes can be responsible for the presence of platelet-leucocyte aggregates (PLA) in the blood [118,119,120]. Circulating PLA is increased in sepsis patients at an early phase, but significantly, decreases in patients developing multiple organ failure [121,122]. Although causality is yet to be demonstrated, it is probably an indirect sign of peripheric sequestration of PLA in the vessel. The role of PLA in determining organ failure is reflected by the complete reversal of acid-induced acute lung injury in a murine model where platelet-neutrophil aggregation was blocked [123].

Platelet-neutrophil aggregates can also potentiate thrombocytopenia through the release of platelet-activating neutrophil extracellular trap (NETs) [124]. NETs formation is critical for the development of sepsis-induced intravascular coagulation, regardless of the inciting bacterial stimulus (gram-negative, gram-positive or bacterial products) [125,126,127].

The role of platelets in supporting the host defense is supported by several studies. It has been shown that thrombocytopenia impairs host defense during Streptococcus pneumoniae infection [128], in Gram negative pneumonia and sepsis [129] and amyloidosis [130]. Recently, a cause—effect relationship was described. In a landmark paper by Claushuis et al., a distinct whole-blood leukocyte transcriptome pattern revealed decreased leukocyte adhesion, diapedesis and extravasation signaling in severe thrombocytopenia [131].

In the last years, while the effect of dysregulated platelet activation has been described, its therapeutic approach is far to be elucidated. Recently, some RCTs have tried to investigate the effects of antiplatelet drugs in septic patients, with controversial results [132,133,134,135,136]. The complex dynamic behavior of platelets during severe infection can be partially responsible for these results. A deeper analysis of the timing of platelet activation can support targeting the antiplatelet therapy before organ failure is developed. Conversely, in patients with already acquired platelet dysfunction, a therapeutic approach aiming to minimize the risk of bleeding can be recommended, and some authors suggest a higher threshold for platelet transfusion in these patients [32]. Recent studies have suggested that platelets dysfunction may result in an “exhausted state,” resulting in an increased risk of bleeding complications [99].

## 7. Conclusions

In conclusion, the role of platelets as a contributor to the immune system has been established. The occurrence of platelet dysfunction during severe infection is common and can be associated with a weaker host response to infections and worse outcomes. Exacerbated platelet activation in sepsis may also contribute to a dysregulation of the inflammatory and immune response in sepsis, which could favor the genesis of multi-organ failure. Assessment of platelet function during infection is gaining relevance as a clinical and research tool useful to improve our knowledge regarding disease pathogenesis and therapeutic options.

## Figures and Tables

**Figure 1 cells-11-00424-f001:**
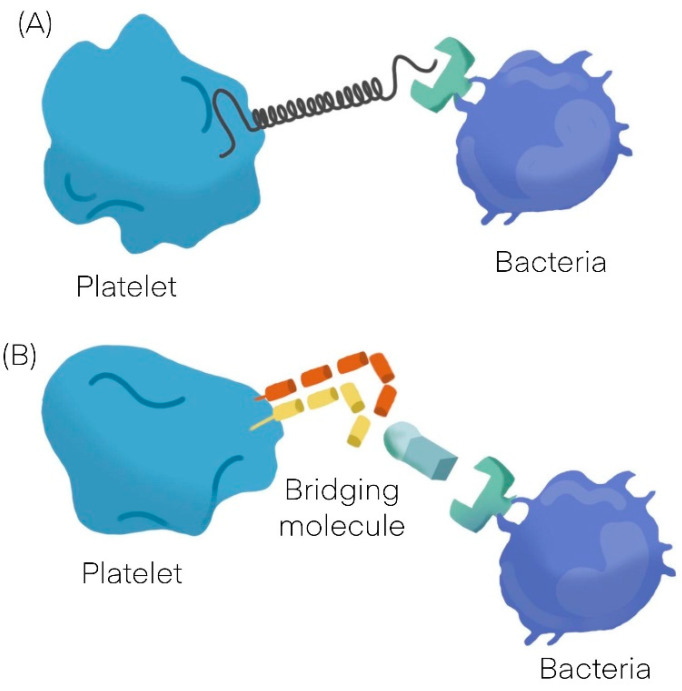
General mechanisms observed in platelet bacterial interaction. (**A**) Direct interaction. (**B**) Indirect interaction.

**Figure 2 cells-11-00424-f002:**
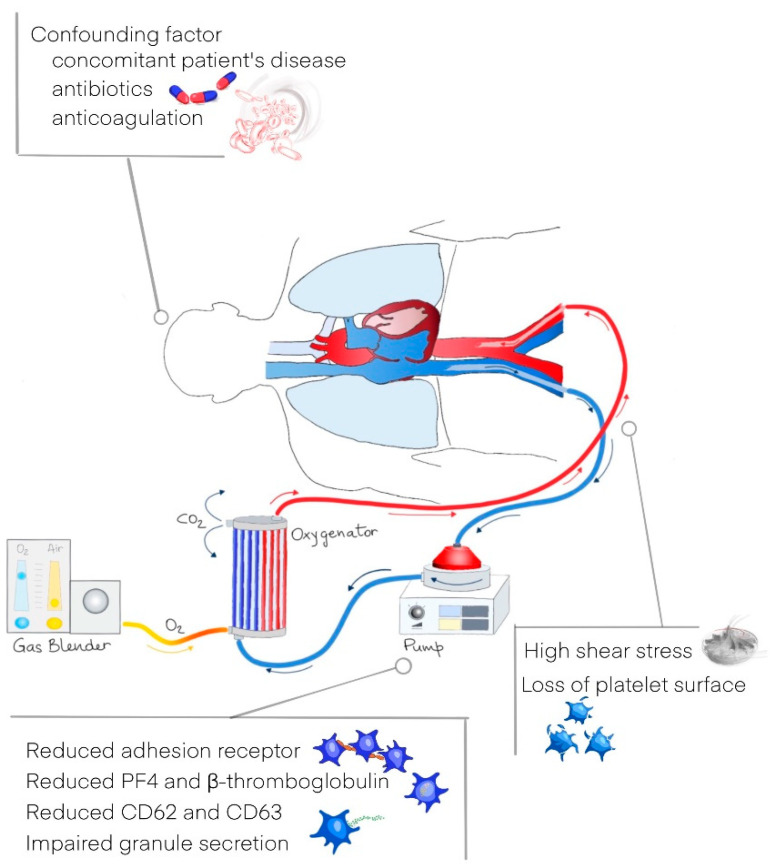
Proposed mechanisms for acquired platelet dysfunction during extracorporeal membrane oxygenation.

**Table 2 cells-11-00424-t002:** Main techniques to monitor platelet function.

Technique	Function	Strengths	Pitfalls
LTA	Evaluation of changes in transmission of light through a sample of platelet-rich plasma (PRP) or platelet suspensions in buffer in response to the addition of a platelet agonist	Less influenced by platelet count Available guidelines on how to interpret LTA results	Time-consuming and technically challenging technique High cost of reagents and consumables
IA	Calculation of the increase in electrical resistance between two electrodes immersed in a diluted sample of whole blood, PRP, or platelet suspension	Shortening of the time window to surgery following P2Y12 inhibitor discontinuation	Misdiagnose of dense granule secretion defects Inferior to LTA for the detection and discrimination of mild platelet function disorders
lumi-aggregometer	Different version of LTA, quantifying the ATP secretion with a luciferin/luciferase assay in parallel with platelet aggregation measures	Information on platelet secretion in addition to platelet aggregation measures	Few reports in the literature on its performance and validation Affected by several variables (concentration of luciferin/luciferase, agonists and ATP standard, volume of PPP and PRP, duration of incubation and measurement, adjustment of platelet count of the PRP).
PFA-200	Assessment of platelet deposition and thrombus growth by microscopy requiring blood to flow over a surface coated with a thrombogenic substrate	Comprehension of the behavior of platelets under physiological and pathological flow, as it occurs within a vessel	Fairly insensitive for the detection of mild platelet function defects
Flow cytometry	Analysis of the expression of activation markers on platelets surface	A smaller volume of blood is needed without platelet-rich plasma preparation	Further validation and standardization tests are required before its application in diagnostic laboratories

LTA: light transmission aggregometry; IA: Impedance aggregometry; PFA: Platelet Function Assay; PRP: platelet-rich plasma; PPP: platelet-poor plasma.

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
