# Peer review of "The Underestimated Role of Platelets in Severe Infection a Narrative Review"

_cells, 2022, doi:10.3390/cells11030424_

Round 1
Reviewer 1 Report
The authors propose an original review that focuses on the interactions between pathogens and platelets and its consequences. This original approach deserves to be published and will be of interest to readers aware of the potential role of platelets in the pathophysiology of sepsis.
However, I believe some elements could be improved before the manuscript is accepted.
In the paragraph on “Platelets interactions with Bacteria”, the text and table 1 do not go into more detail about the receptors that can interact with bacteria. The references of the articles citing these interactions in Table 1 would also be interesting. The interactions according to the type of bacteria or endotoxin could be developed further.
The authors conclude that the development of platelet dysfunction during severe infection is common and may be associated with a weaker host response to infections and a poorer outcome. This point needs to be qualified in the manuscript by the fact that exacerbated platelet activation in sepsis may also contribute to a dysregulation of the inflammatory and immune response in sepsis, which could also contribute to multi-organ failure.
The English language also needs to be reviewed. For example:
P1L13 : please replace by “indeed, sepsis-induced thrombocytopenia can be associated with, or even anticipated by, several changes, including altered morphological pattern, receptor expression and aggregation.”
P1L30 please replace by “Although the main role of platelets is haemostasis, more recently, more attention has been focused on the role of platelets in the host response to infection.”
P1L33. A dot is missing at the end of the sentence.
P10L377 Although causality is yet to be demonstrated
Reviewer 2 Report
In their review article, Fogagnolo et al summarize important findings on platelet activation in bacterial and viral infections. Furthermore, the influence of drugs on platelet function as well as the complications caused by the use of medical devices are discussed and the common platelet function tests are summarized. The review deals with an important timely topic of high clinical relevance. Although most of the key themes are outlined, the review would benefit from more detail in some places (see my exact comments).
Major comments:
- Platelet-pathogen interactions are highlighted in sections 1 and 2, mainly from the point of view of the negative influence of platelets on disease progression after infection. Here, it would be important to also mention the positive effects of platelets in host defense (which are only briefly addressed in the discussion of the current version of the manuscript). For example, platelets can support clearing of circulating bacteria (Wong et al. Nat. Immunology (2013) PMID: 23770641) which also influences the nature of the immune response (Verschoor et al. Nat. Immunology (2013) PMID: 22037602). Furthermore, platelet-pathogen interaction can also locally contain bacteria and retard their distribution witin the organism (Nicolai et al. Nat. Commun. (2020) PMID: 33188196). These aspects should be integrated and discussed in the light of the accepted concept of immunothrombosis, which provides a conceptual framework explaining the transition of a physiological defense mechanism into its host-damaging overreaction (Stark et al. Nat Rev Cardiol (2021) PMID: 33958774).
- due to the timeliness with respect to ChAdOx1 nCoV-19 vaccine, Section 5 would benefit from a short paragraph on platelet activation following vaccination
Minor comments:
- 202:”to reproduce vessel wall”: do you mean “to mimic shear stress from the blood on the vessel”
- Table 2: please correct formatting errors
- 297-300: unclear, you may want to rephrase this sentence
- 356-357: please define T1 and T2
- 378: typo, remove "should be yet"
Author Response
"Please see the attachment

Round 2
Reviewer 2 Report
The authors have addressed all relevant issues. This is a comprehensive and timely review, and I recommend publication in Cells.
This manuscript is a resubmission of an earlier submission. The following is a list of the peer review reports and author responses from that submission.